# Estimating oceanic physics-driven vertical velocities in a wind-influenced coastal environment

Maxime Arnaud<sup>1</sup>, Anne Petrenko<sup>1</sup>, Jean-Luc Fuda<sup>1</sup>, Caroline Comby<sup>1</sup>, Anthony Bosse<sup>1</sup>, Yann Ourmières<sup>1</sup>, and Stéphanie Barrillon<sup>1</sup>

<sup>1</sup>Aix Marseille Univ., Université de Toulon, CNRS, MIO, 13288, Marseille, France

Correspondence: Maxime Arnaud (maxime.arnaud@mio.osupytheas.fr)

**Abstract.** Despite the challenge of measuring them due to their small intensities, oceanic vertical velocities (W) constitute an essential key in understanding ocean dynamics, ocean-atmosphere and biogeochemistry interactions. Coastal events and fine-scale processes (1-100 km / days to weeks) can lead to high-intensity vertical velocities. Such processes can be observed in the Northwestern Mediterranean Sea. In particular, the Gulf of Lion is a region prone to intense north-westerly and easterly wind episodes that strongly impact the oceanic circulation. The JULIO mooring (JUdicious Location for Intrusion Observation) is located on the boundary of the Eastern side of the Gulf of Lion's shelf at the 100 m isobath. JULIO provides Eulerian measurements of three-dimensional current velocities over two main time-periods: 2012-2015, and since 2020. Vertical velocities measurements from JULIO show a good agreement with two independent methods, a Free-Fall Acoustic Doppler Current Profiler and an innovative Vertical Velocity Profiler. To measure physics-driven vertical velocities, we developed a method to identify and filter out biology-induced vertical velocities. Combining satellite and in situ observations with wind model outputs, we identify wind-induced downwelling and upwelling events at JULIO associated with physics-driven vertical velocities with maximum amplitudes of -465/127 m  $day^{-1}$ . Hence, this analysis underlines the need for long term multimethod observations in such coastal areas forced by intense wind episodes. This work presents mooring ADCPs as reliable tools for physics-driven W measurements, with an adaptive algorithm which is applicable anywhere offshore in the ocean to detect W in fine-scale processes.

#### 1 Introduction

Regardless of their presence in most ocean dynamic processes, vertical velocities (hereafter referred as W) remain one of the most complex aspects of today's oceanography. With intensities usually of several orders of magnitude lower than those of horizontal currents, W have been roughly characterized or approximated in the past decades. A breakthrough of (sub)mesoscale resolving models and measurement methods allowed for a better understanding of their dynamics and paved the way for studies combining a wide range of methods. These oceanic vertical velocities can play a key role in fine-scale (1-100 km / days to weeks) physical processes, such as (sub)mesoscale fronts, eddies and others, as detailed in Mahadevan and Tandon (2006) using a wind forced upper ocean circulation model. Moreover, it has been shown that submesoscale physics have a definite impact on oceanic phytoplankton distribution and primary production through different fine-scale features. For example, combining glider

measurements and altimetry-derived dynamic topography through quasigeostrophic theory, Ruiz et al. (2009) estimated large-scale vertical motions linked to chlorophyll transport in the Alboran Sea. A complete summary of submesoscale mechanisms, their induced vertical velocities, and impact on phytoplankton primary production has been carried out by Mahadevan (2016), and their impact on primary export pathways is reviewed in Siegel et al. (2023).

30

Various methods have been developed over the decades to estimate W. The most widely used method features current profilers such as Acoustic Doppler Current Profilers (ADCPs). These ADCPs can be used in different ways, namely mounted on the hull of a vessel or on a mooring line, descended while tied up to a sampling carousel, or mounted on autonomous underwater gliders. In the Gulf of Mexico, D'Asaro et al. (2018) measured  $10^{-2}~m~s^{-1}$  W using an upward-looking ADCP attached to a neutrally buoyant Lagrangian float following three-dimensional movements of water. The same method used by Tarry et al. (2021) in the Alboran Sea exhibited downward vertical velocities up to  $10^{-2}~m~s^{-1}$  versus upward vertical velocities of  $\approx 10^{-3}~m~s^{-1}$ . Lowered ADCP observations also allowed Thurnherr (2011) to measure the W of a few  $10^{-2}~m~s^{-1}$  reaching a precision of  $\approx 5 \times 10^{-3}~m~s^{-1}$ .

Apart from acoustic measurements, glider measurements allow estimating W as the differences between the glider vertical velocities (velocities derived from pressure variations) and their theoretical velocities extracted from a flight model. In the Labrador Sea, Frajka-Williams et al. (2011) estimated W values of  $9 \times 10^{-3}~m~s^{-1}$  in the stratified water column and  $2.1 \times 10^{-2}~m~s^{-1}$  in the mixed layer using gliders. Frajka-Williams et al. (2011) were able to estimate the measurement error of W from gliders respectively at  $5 \times 10^{-3}$  and  $4 \times 10^{-3}~m~s^{-1}$ . With other 3-month observations in the same area by Clément et al. (2024) measured, using gliders, downwelling (upwelling) convective plumes of  $-4.6 \times 10^{-2}~m~s^{-1}$  (3.2 ×  $10^{-2}~m~s^{-1}$ ).

Several other methods are used for W estimation, including in situ observations and theory. In Christensen et al. (2024), Argo floats were used to estimate values of W of a few  $10^{-6}~m~s^{-1}$  using the vertical isotherm displacements over time, where Argo floats are parked at 1000 m. Divergence calculations based on Lagrangian drifters have been carried out for decades. Molinari and Kirwan (1975) first conducted such analyses in the Caribbean Sea computing differential kinematic properties (DKP) of the flow, based on observations of expansion yields inside drifter's clusters. This first method paved the way for multiple studies (Richez, 1998; Righi and Strub, 2001; Spydell et al., 2019) and others. In the Alboran Sea, Tarry et al. (2022) combined it with the continuity equation by making the assumption of null W at the surface. This work highlighted that W could vary, within a four-hour time window, from  $-1.2 \times 10^{-3}$  to  $0.5 \times 10^{-3}~m~s^{-1}$  (respectively -100 and  $50~m~day^{-1}$ ). Esposito et al. (2023), using divergence from drifters as well, obtained W with orders of magnitude around  $1.2 \times 10^{-3}~m~s^{-1}$  (100  $m~day^{-1}$ ). Using electromagnetic APEX (Autonomous Profiling EXplorer) floats and satellite observations, Jakes et al. (2024) measured W with an order of magnitude of  $10^{-3}~m~s^{-1}$  (>100  $m~day^{-1}$ ) by sampling a cold filament in the region of the Antarctic Circumpolar Current. They used the same treatment method as Phillips and Bindoff (2014) who obtained, in the same region near Kerguelen, a negative depth-averaged W around  $10^{-5}~m~s^{-1}$  and an associated standard error  $\approx~10^{-4}~m~s^{-1}$ .

Numerical estimations of W led Garcia-Jove et al. (2022) to combine model simulations with UCTD casts and glider measurements in Alboran Sea frontal regions, thus reaching values with an order of magnitude of  $10^{-4} \ m \ s^{-1}$  (55  $m \ day^{-1}$ ). Another numerical approach for W estimation relies on the Q-vector version of the  $\omega$ -equation (Hoskins et al. (1978), adapted by Tintoré et al. (1991); Pollard and Regier (1992)) applied in quasigeostrophic theory. Oceanic vertical velocities are thus

expressed through the adiabatic QG  $\omega$ -equation. The numerical solving of the  $\omega$ -equation, using three-dimensional maps of density and horizontal velocities (Fiekas et al., 1994; Giordani et al., 2006; Hoskins et al., 1978; Rudnick, 1996), provided W with an order of magnitude from  $10^{-5}$  to  $10^{-4}$  m  $s^{-1}$ . With the same method, Zhu et al. (2024) obtained W using horizontal currents from a ship-mounted ADCP, reaching an order of magnitude of  $10^{-3}$  m  $s^{-1}$  (170 m  $day^{-1}$ ) in the Kuroshio-Oyashio extension region.

65

Regarding other theory based estimations, Cortés-Morales and Lazar (2024) analyzed Linear Vorticity Balance (LVB) from an eddy-permitting ocean circulation model to estimate W and investigate their interannual variability on large spatial scales. By integrating the geostrophic component of LVB vertically, and average it over 56 years, they estimated W values with an order of magnitude up to  $10^{-5}$  m  $s^{-1}$  giving us an idea of the intensity of W at large space and time scales. Alternatively, relationships between surface variables (i.e. horizontal velocities, surface density and derived variables) and inferred vertical velocities have been evaluated through different machine learning models by He and Mahadevan (2024).

Our area of study is located in the Northwestern Mediterranean Sea, which exhibits a large scale cyclonic circulation (Millot, 1999). In the Northwestern basin, the Northern Current (NC) constitutes its Northern branch (Millot and Taupier-Letage, 2005). Originating from the Ligurian Sea, this density current follows the coast with a horizontal speed from 0.4 up to 0.7  $m s^{-1}$  in winter, with a width, depth and flow rate that are prone to seasonal variability (Petrenko, 2003). The Northwestern basin is a microtidal environment; therefore, the tidal effects are negligible in this study. With regard to atmospheric forcing, the area is prone to intense wind episodes from two main directions: easterly and north-westerly winds. These forcings and their impact on the oceanic circulation have been studied (Guenard et al., 2005; Berta et al., 2018; Barral, 2022). These winds exhibit a strong seasonal variability that affects both their direction and speed. Models (Lebeaupin Brossier et al., 2013), reanalysis (Soukissian and Sotiriou, 2022) and observations (Ragone et al., 2019) showed that the so-called Mistral wind (Northwestern wind) strengthened (intensities between 15 and 20  $m s^{-1}$ ) during the cold seasons and weakened during warmer ones (between 5 and 10  $m s^{-1}$ ), with rare intensity peaks. Strong Mistral episodes enabled deep convection in winter (Testor et al., 2018), associated with non-negligible W (Bosse et al., 2021; Giordani et al., 2017). The Northwestern basin is also subject to numerous fine-scale processes (d'Ovidio et al., 2010, 2019; Barrier et al., 2016; Declerck et al., 2016). Wind directions also have a direct impact, via Ekman transport, on water masses moving toward or away from coasts, thus generating, respectively, downwelling or upwelling events (Barrier et al., 2016; Mourre et al., 2022). As tidal effects are negligible (Lamy et al., 1981), these upwellings and downwellings are mainly induced by the occurrence of specific wind events, the presence of coasts and of the NC.

The vertical measurements presented before occurred in both coastal and offshore environments. A few estimations of vertical velocities have been performed around our area of study (the Gulf of Lion and surroundings). Lowered ADCPs paved the way for a new method adapted by Comby et al. (2022) in the Ligurian Sea. Comby et al. (2022) cross-analyzed different observations using ADCPs, with a classical four-beam ADCP and a new generation five-beam ADCP both lowered on carousels, and implemented a new method with a four-beam ADCP free-fall deployment. These methods allowed for measurements of W about several  $10^{-3} m \, s^{-1}$ , with standard deviations of the same order of magnitude. In the Gulf of Lion, Margirier et al. (2017) studied convective events. They compared glider W measurements to ADCP measurements from the LION mooring line (43°

N;5° E) and obtained W of a few  $10^{-2}~m~s^{-1}$ ; while the measurement error of W from gliders is evaluated at  $5 \times 10^{-3}$  and  $4 \times 10^{-3}~m~s^{-1}$  (Merckelbach et al., 2010). Based on the principle of glider flight model, Fuda et al. (2023) have proposed a new approach to measure W: the Vertical Velocity Profiler (VVP). The VVP allowed vertical velocity measurements with orders of magnitude of a few  $10^{-3}~m~s^{-1}$  in the region of the North Balearic Front. In the Ligurian Sea, the  $\omega$ -equation was calculated from ADCP horizontal velocities and densities derived from a moving vessel profiler, leading to estimated vertical velocities of  $10^{-4}~m~s^{-1}$  (Rousselet et al., 2019). The same method was used by Tzortzis et al. (2021) in the South Balearic Islands frontal region where W were in the order of a few meters per day.

100

110

Generally, physics-driven W have very low intensities. Nonetheless, some W can be strong when driven by biological processes such as diel vertical migrations (DVM). DVMs have been analyzed for decades (Enright and Honegger, 1977; Haney, 1988), highlighting both diurnal and seasonal cycles. Heywood (1996) estimated W of several centimeters per second associated with living scatterers using a vessel-mounted ADCP. DVMs have also been observed with moored upward-looking ADCPs in different regions: Alboran Sea (van Haren, 2014), Corsica Channel (Guerra et al., 2019) and Norwegian Sea (Cisewski et al., 2021). Cisewski et al. (2021) identified specific oceanic and optical conditions as important drivers for DVMs while Guerra et al. (2019) linked DVMs to Chlorophyll-a peaks. van Haren (2014) also showed the direct impact of ocean stratification and internal waves on living scatterers behavior and movements. The behaviour of zooplankton swarms has been studied using an association of echosounder and ADCP data (moored Sourisseau et al. (2008), ship-mounted Tarling et al. (2018)). The size of living scatterers also impacts the time at which vertical migrations take place and the scatterers moving speed (Gastauer et al., 2022). In the Strait of Gibraltar, a recent study coupling an upward-looking ADCP and an echosounder has highlighted the impact of DVMs on the intensity of horizontal currents (Sammartino et al., 2024).

Between the ascending (descending) phases at dusk (dawn) of DVMs, the zooplankton stays at the surface or subsurface during the night. These stationary patches are sometimes associated with recorded ADCP negative vertical velocities (Tarling et al. (2001), BioSWOT-Med (10.17600/18002392) and FUMSECK (10.17600/18001155) cruises (unpublished work)). Some hypotheses attempt to explain these counterintuitive negative vertical velocities of stationary patches. Among them, a sinking phenomenon due to satiation of living scatterers that occurs during the night as mentioned in Tarling and Thorpe (2017) or the angle of displacement of living scatterers, varying depending on whether they ascend or descend in the water column thus impacting the ADCP returning signal (pers. comm. M. Ohman).

The aim of this study is to measure physics-driven W in a coastal environment and study specific events such as upwellings or downwellings. To estimate the reliability of our data, different measurement methods are compared, giving an insight into our measurement precision. The JULIO (JUdicious Location for Intrusion Observation) mooring ADCP provides in situ time series of 3D currents from 2012 to 2015, and since 2020 (Petrenko et al., 2023). These measurements are completed with observations such as satellite SSH (Sea Surface Height), SST (Sea Surface Temperature), and in-situ temperature measurements. Within the framework of these observations, an analysis of the intensity and variability of W is conducted using the moored ADCP dataset. Our scientific questions are:

- Is the JULIO moored ADCP a reliable tool to measure physics-induced vertical velocities?

- How do the measured vertical velocities relate to complementary observations?
- Can we identify typical coastal dynamical processes?

The paper is structured as follows. In the Data and Methods section, the ADCP dataset is described, followed by datasets obtained with the Vertical Velocity Profiler (VVP) and the Free-Fall ADCP (FF-ADCP) method. The first part of the results section consists in the inter-comparison and validation of W measurements using JULIO ADCP, FF-ADCP and VVP methods. In the second part of the results section, we take a global view of all vertical velocity time series before focusing on a methodology developed to isolate and filter biology-induced signals. This filtering allows to present oceanic dynamics resulting from physical processes in the third part. This latter part outlines the analysis of both physics-induced vertical velocities and surface observations, focusing on four specific cases: two upwelling and two downwelling events. The outcomes are then discussed in terms of the questions raised on both precision and variability, followed by the conclusions and perspectives.

## 2 Data and methods

145

## 140 2.1 Moored ADCP (JULIO)

The JULIO mooring is located offshore of Marseille on the border of the Gulf of Lion shelf, on the 100 m isobath (Fig. 1). JULIO is composed of a RDI (Research, Development and Innovation) 300 kHz Workhorse Sentinel upward-looking ADCP with four classical beams. Before 2020, it was set up in a bottom anti-trawling device; after 2020 on a short mooring line minimizing the risk of trawling. Moored ADCPs provide three-dimensional Eulerian vertical profiles of currents over time, by measuring a shift in frequency between emitted and received sound wave, indicating the velocity of particles suspended in the water, and then transposing these data from a beam reference frame to a geographical one. The method implies the hypothesis that these particles retrodiffusing the acoustic signal are passively drifting in the water, transported by the oceanic currents. For the JULIO ADCP the velocity resolution, hence the minimal achievable value, is  $1 \times 10^{-3}$  m  $s^{-1}$ . Time series are obtained with a time resolution of half an hour and a vertical resolution (i.e. cell size) from 4 m to 2.5 m (Table 1). More information can be found on JULIO's website: https://people.mio.osupytheas.fr/~petrenko/julio.htm. The observations (interrupted by yearly maintenance and trawling incidents) began in 2012 and were suspended from 2015 to 2020. The initial purpose was to measure the Northern Current intrusions on the continental shelf of the Gulf of Lion (Barrier et al., 2016). Time series are shown in Table 1.

The 3D current measurements are obtained after a data-quality treatment inherent to the measurement method. A layer, known as the blanking distance, is removed near the ADCP, as a bias is caused by its transducers ringing. This ringing ceases progressively with time and thereby influences the closest echoes of the ADCP. The thickness of the blanking zone varies between  $\approx 5-6$  m (varying with slight changes of ADCP configuration throughout the years, Table 1). Another layer, known as the Sidelobe interference layer, is removed at the surface since the water/air interface acts as a powerful reflector, and returns

Figure 1. Location of the JULIO ADCP mooring on the averaged December 2022 current horizontal velocities intensity (CMEMS reanalysis)

| Time series | First day  | Last day   | Bin size (m) | ADCP depth (m) |
|-------------|------------|------------|--------------|----------------|
| 1           | 02-12-2012 | 10-23-2012 | 4            | 98             |
| 2           | 09-26-2013 | 03-28-2014 | 4            | 98             |
| 3           | 07-17-2014 | 04-10-2015 | 4            | 101            |
| 4           | 12-07-2020 | 08-31-2021 | 4            | 84             |
| 5           | 09-01-2021 | 06-23-2022 | 4            | 82             |
| 6           | 06-24-2022 | 06-22-2023 | 2.5          | 83             |
| 7           | 07-12-2023 | 05-21-2024 | 2.5          | 84             |

Table 1. JULIO time series with corresponding dates of beginning and ending of data recording, bin size and ADCP depth.

an intense signal to the ADCP. This layer has a thickness of  $\approx 11$  m (respectively  $\approx 16$ -19 m) when the vertical bin size is 2.5 m (respectively 4 m). This layer thickness is more conservative than that calculated as a function of the acoustic beam's slant angle (here 20°), keeping only high-quality data (ADCP's principles of operation, https://www.teledynemarine.com/en-us/support/

SiteAssets/RDI/Manuals%20and%20Guides/General%20Interest/BBPRIME.pdf). The mooring ADCP presents two types of measurement errors: an intrinsic precision linked to the instrument's hardware, and a standard deviation depending on the deployment parameters. The quadratic sum of these precisions gives a conservative error of  $\approx \pm 5$  mm  $s^{-1}$  on W for JULIO.

The ADCP also provides the strength of the returning ADCP acoustic signal (echo intensity, measured in dB) after hitting particles in the water, which decreases with the distance to the target. At a given distance, the higher the echo intensity, the higher the concentration of particles. We computed the Echo Amplitude Anomaly (EAA) in three steps: 1) we select daytime echo amplitude (between 6 a.m. and 6 p.m.) to avoid nocturnal phenomena (Sect. 3.3), 2) we compute a trend by applying a rolling mean with a 15 days time window and 3) we subtract this trend from all echo amplitudes, resulting in EAA.

## 2.2 Vertical Velocity Profiler

165

The Vertical Velocity Profiler (VVP) is an innovative instrument developed at the Mediterranean Institute of Oceanography (MIO) laboratory to measure W with the help of a flight model. Based on the methodology developed for the gliders (Merckelbach et al., 2010; Frajka-Williams et al., 2011; Bosse et al., 2021), W are obtained by comparing the instrument measured velocity (derived from pressure) to its theoretical velocity (obtained with a flight model). The VVP, driven by a propeller to a set depth, rises freely under the sole effect of its positive buoyancy. During the rise, the pressure is measured at 2 Hz, giving the instrument vertical velocity through the water column by calculating the pressure time derivative. This velocity is then compared to the theoretical one based on its flight model. Any difference between these values is considered as upward or downward W (Fuda et al., 2023). The VVP allows autonomous measurements of vertical velocities inside the water column with a sampling frequency of 2 Hz and a vertical resolution depending on the rising speed of the VVP. In our case, the rising speed was  $7.40 \times 10^{-2} m\ s^{-1}$ . As the VVP's wake induces an interfering  $20\ s$  period sine wave component on the W of the VVP, the signal is smoothed to a  $30\ s$  temporal resolution (the signal is low-pass filtered with a cutoff frequency of  $\frac{1}{30}\ Hz$ ). The VVP profile depth range was from 78.9 to 3 m. For this study, we focused on the only VVP rise concomitant with Free-Fall ADCP measurements: on the 06-24-2022 from 9:17 to 9:34 UTC. This time range is the only period matching for the three instruments: after the redeployment of the mooring line at the JULIO point, with the VVP launched  $\approx 15\ \text{minutes}$  after JULIO was released back into the water and the FF-ADCP sampling starting as soon as possible after these two previous operations.

## 2.3 Free-Fall ADCP

The Free-Fall ADCP (FF-ADCP) consists of a downward-looking ADCP mounted on a weighted cage, connected to the boat by a rope which is left loose enough on the descent in order to let the ADCP drop free of the ship's movements. This assures a descent of the FF-ADCP as vertical and smoothly as possible. The cage is then pulled back in order to proceed as quickly as possible to the next descent to measure the temporal variability of vertical velocities at the highest possible frequency. With a descent down to 200 m at a speed of  $0.65 \, m \, s^{-1}$ , eight casts can be performed in 1 hour. Only the top-down profiles are exploited for W measurement, as the pull up carried out with the ship's which introduces large perturbations of the measurement. The work conducted by Comby et al. (2022) presents different data processing methodologies depending

on the type of ADCP data used (W from four beams or fifth beam). The main paper recommendation was the use of the free-fall method with a five-beam ADCP which would most likely provide the best W measurement. In our case, with a new fifth beam ADCP, we follow the data treatment of Comby et al. (2022). Namely, 1) the application of three quality criteria, 2) the generalization of attitude angles (i.e. pitch, roll and heading) in spherical convention, and a correction of the fifth beam deviation with rotation matrices, 3) the removal of the instrument falling speed to the absolute velocity measured and, finally, 4) the 10 seconds temporal smoothing. FF-ADCP data were collected on the same day as both the beginning of the JULIO ADCP 6th time series and the VVP measurements.

## 2.4 Wind model

In addition to these in situ observations, model outputs were used. From the global numerical prediction model ARPEGE (Bouyssel et al., 2022), we extracted the horizontal components of wind velocities at 10 m above the surface, with a horizontal resolution of 0.1°. The model has a temporal resolution of 3 hours. We focused our data analysis at JULIO coordinates (i.e. 43.142° N;5.233° E), and have expanded our study area to the whole Gulf of Lion's frame (41.25° N : 44° N;2° E : 7° E) to discriminate (spatially and temporally) isolated punctual wind peaks from wind episodes that affect the entire region.

## 2.5 Satellite data

215

210 Satellite observations are also analyzed in order to obtain surface data at or surrounding the JULIO mooring.

## 2.5.1 Sea Surface Temperature

Sea Surface Temperature (SST) product, obtained from Copernicus Marine Services, combines in situ data from Canadian Integrated Science Data Management centre (ISDM) and MyOcean In Situ Thematic Centre, with satellite data from MODIS, AATSR, AVHRR, and SEVIRI (Buongiorno Nardelli et al., 2013, 2015). The resulting dataset provides daily gap-free observations and a horizontal resolution of  $1/16^{th}$  degree (Mediterranean Sea High Resolution and Ultra High Resolution Sea Surface Temperature Analysis L4 NRT from satellite observations, E.U. Copernicus Marine Service Information (CMEMS), Marine Data Store (MDS), DOI: 10.48670/moi-00172; accessed on 03-03-2025).

## 2.5.2 Sea Surface Height

Sea Surface Height above sea level (hereafter called SLA for Sea Level Anomaly) dataset gathers observations from SSALTO and DUACS altimetry products, resulting in a daily gridded dataset featuring a horizontal resolution of  $1/8^{th}$  degree (European Seas Gridded L4 Sea Surface Heights And Derived Variables Reprocessed (1993 ongoing) from satellite observations, E.U. Copernicus Marine Service Information (CMEMS), Marine Data Store (MDS), DOI: 10.48670/moi-00141; accessed on 03-03-2025).

## 2.6 In situ Sea Surface Temperature with HTMNet

SST was also measured at 5 coastal stations around JULIO: Carro, Redonne, Cassis, La Ciotat and Le Brusc, located between 5 and 6 degrees of longitude East as part of the HTMNet program (Rey et al., 2020). The temperature is measured since 2019 at the very surface of the water (in the first meter of depth) with a temporal resolution of 2 minutes (HTMNet website: https://htmnet.mio.osupytheas.fr/HTMNET/squel.php?content=accueil.php).

## 3 Results

235

## 230 3.1 Inter-comparison of W measurements

Three W measurements have been made simultaneously in space and time at the JULIO site in 2022: JULIO's mooring ADCP, VVP, and FF-ADCP (Fig. 2). Note that only in this case and to compare our data properly, JULIO profiles have been vertically smoothed (using a rolling mean), to a vertical resolution of 5 meters. All methods show profiles with values varying mainly between  $\pm 1 \times 10^{-2} m\ s^{-1}$  and mean values close to  $0\ (2 \times 10^{-4},\ 0 \times 10^{-4}\ and\ -2 \times 10^{-4}\ m\ s^{-1}$ , respectively for JULIO, VVP and FF-ADCP). In terms of standard deviation, the VVP shows the largest variability with a standard deviation of  $\pm 5.9 \times 10^{-3} m\ s^{-1}$  versus the FF-ADCP with  $\pm 4.1 \times 10^{-3} m\ s^{-1}$  followed by JULIO ADCP  $\pm 1.0 \times 10^{-3} m\ s^{-1}$ . These W are observed under calm conditions (South-South-Westerly wind under  $10\ m\ s^{-1}$  during the previous 24 hours). All the measurements are compatible, taking into account measurement errors of a few  $10^{-3}\ m\ s^{-1}$  and their different intrinsic temporal features.

For each measurement, the standard deviation includes both measurement error and environmental variability. For JULIO, the standard deviation exhibits values mostly under  $\pm 5~mm~s^{-1}$ , below the measurement error ( $\approx \pm 7~mm~s^{-1}$ ) given by the manufacturer. VVP measurement error relies mostly on its comparison with the flight model, giving a few  $mm~s^{-1}$  obtained with a sensitivity analysis. The FF-ADCP method involves two instruments: the pressure sensor and the ADCP. The resulting measurement error presents an order of magnitude of  $\pm 2~mm~s^{-1}$ .

The three different measurements are compatible in means and standard deviations, showing no measurement bias for one instrument with respect to the others. However, it is important to stress that the instruments capture different temporal features (Fig. 2, bottom). As JULIO takes a snapshot of the entire water column every 30 min, the VVP slowly measures the water column during its 17-minute ascent. The FF-ADCP measures the water column during 2 to 3 min while moving downward. The variability observed by the VVP and FF-ADCP is likely due to the presence of internal waves.

## 250 3.2 Vertical velocities on a yearly scale

All vertical velocities measured with JULIO, for all the time series and depths, are represented as yearly probability density functions (PDF) with a kernel smoother (Fig. 3). Annual sampling varies from one year to another regarding the duration, the number of months, and the sampled seasons (Table 1). The number of observations for each year is between  $1.7 \times 10^4$  and  $3.98 \times 10^5$ . The PDFs exhibit, for all cases except 2013, a negative skewness with a mean consistently below the median value

Figure 2. W profiles  $(m \ s^{-1})$  as a function of depth (m). Up, from left to right: JULIO (one snapshot every 30 min), VVP (single profile lasting 17 min), and FF-ADCP (profiles lasting 2 to 3 min in a row). In blue, mean (solid line) and standard deviation of the profiles. The profiles are colored-mapped as a function of time (June 24th 2022 from 8:30 a.m. to 10:00 a.m. (UTC)). Down: W  $(m \ s^{-1})$  as a function of depth and time for the three methods.

which is equal to zero for all years except 2021 and 2022. Boxplots highlight that 50% of the values have an order of magnitude around  $\pm 5 \times 10^{-3} \ m\ s^{-1}$ . Nonetheless, one should note that although the sampling frequency remains the same, the number of months sampled varies depending on the year.

## 3.3 Negative vertical velocities at night-time

260

265

The signal obtained with JULIO ADCP data exhibits an intense and recurring pattern of negative W at night (example shown in Fig. 4), at the origin of the negative skewness observed in the general PDFs, with specific characteristics. Strong negative vertical velocities appear in patches (with averaged W in the patches  $= -1.8 \times 10^{-2} \ m\ s^{-1}$ ), describing a diurnal cycle spanning  $\approx$ 8 hours and centered at midnight. Those patches are located mainly under the surface (between the surface and  $\approx$ 50 meters depth) with a seasonal variability and a more pronounced presence in springtime (Fig. 5).

With these characteristics, our assumption is that the origin of this signal is biological. As explained in the introduction, this corresponds to patches of zooplankton, stationary at night at the surface or subsurface, associated with negative vertical

**Figure 3.** Kernel density estimations of vertical velocities at all depths by year of observation. Scatters: overlapping raw vertical velocity values. Boxplot: whiskers represent extreme values while boxes show the first and third quartiles, thus containing the middle 50% of the values. The black line inside the box represents the median and the red cross, the mean. The number of observations for each year can be found on the right.

velocities. In terms of order of magnitude and despite different spatial scales and species, the same type of signals were observed with a 153 kHz ADCP in Tarling et al. (2001) where nighttime W reached intensities up to  $\approx -5 \times 10^{-2}~m~s^{-1}$ . Regarding descending phases of DVMs, measurements of W in the Northeast Atlantic (Heywood, 1996) exhibit values of downward swimming speed with the same order of magnitude: between 2 and  $6 \times 10^{-2}~m~s^{-1}$ . We use the EAA, derived from the measured echo intensity, to help the identification of these patches (Fig. 6 top panel). This conjunction of high-intensity patches helps to strengthen the biological signal hypothesis.

**Figure 4.** Vertical velocities as a function of depth and time (month-day/hour) over 11 days (April to May 2023). Pink and orange lines show the 19.5 and 75.5 m depth, respectively. The vertical red dashed lines mark midnight while the dotted ones mark noon.

**Figure 5.** Monthly Probality Density Functions for all the time series, during night time (from 8PM to 4 AM), at shallow and bottom depths (respectively 20 and 75 m; the two different layers referring to the pink and orange lines of Fig. 5). Top: vertical velocities. Down: Echo amplitude anomaly. The colored dots show the number of observations (top and bottom together).

## 3.4 Patches identification

As targeted patches are characterized by intense W and EAA, their identification relies on thresholds set on both variables. In Fig. 5, we identified negative skewness of W PDF during spring and summer nights in the upper layer, beginning in April,

Figure 6. Top to bottom. (a) Echo amplitude anomaly (EAA) as a function of depth and time (month-day/hour) over 11 days (April to May 2023). (b) Vertical velocities with intensities lower than  $-7.0 \times 10^{-3} \ m \ s^{-1}$ . (c) Tagged data with  $W < -7.0 \times 10^{-3} \ m \ s^{-1}$  & EAA > 0, black boxes represent the patch criterion. (d) Patches identified as biological signal.

with a median at  $-7.0 \times 10^{-3}~m~s^{-1}$ , which will be our W threshold. The top-layer EAA features a strong positive skewness that perfectly matches our W negative anomaly. We therefore consider the combination of W below the threshold of  $-7.0 \times 10^{-3}~m~s^{-1}$  and EAA above 0, as the initial step to identify biology-induced signal.

A study in Scotia Sea (Tarling et al., 2009) exhibited different types of swarm and their associated characteristics. Their thicknesses can vary from 2 to  $\approx$ 30 meters, the latter value corresponding to super swarms described as relatively rare in the study. The study also highlights that these phenomena last  $\approx$ 8 hours. Nevertheless, as our area of study differs from theirs, we conducted a sensitivity analysis leading to optimized characterization in time and depth of these events. Therefore, we chose to create a virtual box with dimensions of  $\approx$  12 meters depth and 4 hour long. The method consists in moving the box through the entire dataset over time and depth as shown in Fig. 6 panel c). Inside each position of the box, data are tagged if more than 70% pass both W and EAA thresholds. The tagged data are shown in Fig. 6 d) and thereafter called biological signal. These tagged data are then filtered out from the W dataset, resulting in the physics-driven W, shown in Fig. 8.

Figure 7. Depth average (over the whole water column) of W during all the time series. The orange (blue) scatters are positive (negative) values of W, and the green scatters represent what is identified as the biological-induced signal, afterwards filtered out. The orange and blue boxes are respectively our targeted upwellings and downwellings. The black horizontal lines correspond to  $\pm 5 \ mm \ s^{-1}$  thresholds.

## 3.5 Overview of W time series

W measured with the ADCP at JULIO show a general variability of  $5.0 \times 10^{-3}~m~s^{-1}$  around  $0~m~s^{-1}$  (Fig. 7) with 50% of the values between -5 and  $5 \times 10^{-3}~m~s^{-1}$  in Fig. 3. This variability can also be seen in the depth-averaged W global view over all available JULIO time series (Fig. 7). The tagged biological signals show a seasonal pattern: they occur intensely from April to June, and sometimes extend from March to October with less occurrences. Events showing positive or negative W intensities beyond this variability are nonetheless frequently observed.

## 3.6 Events occurrences

East of the Gulf of Lion, the main wind forcings are Mistral, which refers to north-westerly winds, and easterly winds. The Mistal (respectively easterly winds) induce Ekman dynamics and important cross-shelf water transports offshore (onshore) causing upwellings (downwellings) strongly linked to seasonality (Millot and Wald, 1981; Barrier et al., 2016).

This work focuses on upwelling and downwelling events. Once the biological-induced signal is filtered, physics-driven events identification follow four criteria in the following order. We first identify groups of physics-driven vertical velocities with intensities greater than  $\pm$  10 mm  $s^{-1}$  lasting at least 2 hours (5 values) and spreading deeper than  $\approx$  10 m. If the criterion is met, we look for groups of horizontal velocities in the proximity of the identified W, that match both intensities greater than  $\pm$  200 mm  $s^{-1}$  and offshore (onshore) directions, consistent with an upwelling (downwelling) event. If these two criteria are met, we analyze wind speed intensity and direction from ARPEGE model, keeping episodes where intensities are over  $10 \ m\ s^{-1}$  (Berta et al., 2018) and directions are north-westerly (south-easterly) to match offshore (onshore) horizontal currents. Finally, this whole identification is completed by an analysis to detect the expected variability in SST (SLA) satellite observations leading to the upwelling (downwelling) event identifications.

This method has identified a dozen of events with 9 upwellings and 3 downwellings. We chose two upwellings and two downwellings after a global analysis of all time series, as the most intense and consistent events. The targeted upwellings (downwellings) occurred around the beginning of spring (winter) and are described in the following.

## 3.6.1 Upwellings

An upwelling  $(U_{2022})$  with intense positive W was detected at JULIO, after biology filtering, from April 1st to April 3rd 2022 (Fig. 8, right panel). W averaged over depth and time during these two days has a positive value of  $1.5 \times 10^{-3} \ m \ s^{-1}$  (versus  $-3.9 \times 10^{-3} \ m \ s^{-1}$  before biology filtering) which amounts to  $\approx 127 \ m \ day^{-1}$ . To estimate the sensitivity of the filtering method to its parameters, we compute the same W average over  $U_{2022}$  for two different sizes of boxes, keeping the same thresholds (W<-7 × 10<sup>-3</sup>  $ms^{-1}$ , EAA>0 and a criteria of 70%). With the smaller box (8 meters depth and two hours), W is  $2.6 \times 10^{-3} \ m \ s^{-1}$  ( $\approx 227 \ m \ day^{-1}$ ). With the bigger box (20 meters depth and eight hours), W is  $-0.2 \times 10^{-3} \ m \ s^{-1}$  ( $\approx -16 \ m \ day^{-1}$ ). The most likely explanation is that the smaller box identifies negative W that are not biology-induced as the bigger box probably misses some of the biology-induced signals. The horizontal velocity components correspond to a current heading

Figure 8. Top to bottom (4 panels): zonal velocity (U), meridional velocity (V), vertical velocity (W), as a function of depth and time (month-day/hour), wind intensity and direction; all during a 5 days window. Last panel: SST observations over 2 months (month-day) at JULIO (dashed red line for JULIO nearest SST point, solid black line for an average on  $\Delta lat = 0.14$  and  $\Delta lon = 0.18$  centered on this point), and SST temporal gradient (colored dots). The left and right panels show the two different upwellings:  $U_{2012}$  (left) and  $U_{2022}$  (right), each indicated between the two vertical magenta lines.

offshore for the entire water column. Such flow matches an intense north-westerly wind episode with speeds exceeding 20  $m \, s^{-1}$  (Fig. 8).

Considering satellite observations, analyzed SST (Fig. 8) shows a drop in temperature between April 1st and April 3rd before a general increase according to seasonal trends. The SST decreases from 13.65°C on April 1st, to 12.75°C on April 3rd (0.9°C

Figure 9. Top to bottom (2 panels): model wind direction and intensity in  $m \ s^{-1}$  as a function of time (year-month-day), temperature measured at HTMNet stations in °C for all stations (black line mean value and grey standard deviation) and at the nearest station to JULIO (green).  $U_{2022}$  is indicated in both panels between the two vertical magenta lines.

drop). Over a two-month period (Fig. 9), in situ temperature data show a singular temperature drop (2°C) matching the most intense north-westerly wind episode.

Another upwelling  $(U_{2012})$  occurred 10 years before, in March 2012 (Fig. 8, left panel). Though the configuration differs by much fewer biology-induced patches, the same features are observed: strong north-westerly wind, south-eastward current and a steep change in W intensity. The wind episode was just slightly shorter than during the upwelling previously discussed. The W signal varies during this mainly positive period, with a variability decrease as wind intensity drops below  $15~m~s^{-1}$ , which appears to be a common threshold in these two events. The intense positive W during the upwelling lasts from March 5th to March 7th. The depth-time averaged W during this period is  $1.1 \times 10^{-3}~m~s^{-1}$ , leading to  $\approx 95~m~day^{-1}$ . Observations of SST show a small decrease of  $\approx 0.5$ °C leading to a value which remains constant (12.8°C) for two days. Note that  $U_{2012}$  happened before HTMNet SST observations were available.

## 3.6.2 Downwellings

325

335

An intense (around  $10~m~s^{-1}$ ) easterly wind episode occurred during almost 2 days at the end of fall 2014 (Fig. 10, left panel). This event is followed by a shift in the wind direction and a decrease in its intensity, leading to a weak southerly wind. During this decrease, strong positive meridional component of velocity reflects an onshore horizontal current, leading to a downwelling  $(D_{2014})$ . The latter features strong negative W with a depth-time average during the event (12 hours) of  $-5.4 \times 10^{-3}~m~s^{-1}$  which is equivalent to  $\approx$ -465  $m~day^{-1}$ . Surface observations of SLA (Fig. 10) show an increase during November, possibly linked to these easterly winds.

A second downwelling ( $D_{2021}$ ) taking place in December 2021 also exhibits negative W (Fig. 10, right panel) following a 24-day long easterly wind episode. Horizontal currents show a strong variability during and after the wind episode. The

Figure 10. Top to bottom (4 panels): zonal velocity (U), meridional velocity (V), vertical velocity (W), as a function of depth and time (month-day/hour), wind intensity and direction (5 days window). Last panel: SLA observations over 2 months (month-day) at JULIO (dashed red line for JULIO nearest SLA point, solid black line for an average on  $\Delta lat=0.14$  and  $\Delta lon=0.18$  centered on this point), and SLA temporal gradient (colored dots). The left and right panels show the two different downwellings:  $D_{2014}$  (left) and  $D_{2021}$  (right), each indicated between the two vertical magenta lines.

time-depth averaged W during the event (12 hours) indicates a value of  $-3.9 \times 10^{-3} m \ s^{-1}$ , equivalent to -346  $m \ day^{-1}$ . At the same time, SLA is reaching a local maximum after a one week rise (amplitude  $\Delta h \approx +0.05 \ m$ ).

## 3.7 Global view on these events

With no temporal average, depth averaged W values exhibit maximum values around  $2.55 \times 10^{-2}~m~s^{-1}$  for  $U_{2012}$  and  $U_{2022}$ , versus minimum values around  $-1.93 \times 10^{-2}~m~s^{-1}$  for  $D_{2014}$  and  $D_{2021}$ . The depth averaged W (Fig. 7) exhibit maximum values of  $8.5 \times 10^{-3}~m~s^{-1}$  (respectively  $7.7 \times 10^{-3}~m~s^{-1}$ ) during  $U_{2012}$  (respectively  $U_{2022}$ ). Regarding downwellings, minimum values were detected at  $-6.5 \times 10^{-3}~m~s^{-1}$  (respectively  $-5.7 \times 10^{-3}~m~s^{-1}$ ) during  $D_{2014}$  (respectively  $D_{2021}$ ). Noticeable events such as upwellings and downwellings are visible outside the general variability of  $\pm 5.0 \times 10^{-3}~m~s^{-1}$  (Fig. 7).

Our observed upwellings show southeast horizontal currents lining up on the path of the north-westerly winds over the entire water column. Note that the classic offshore Ekman direction of the horizontal currents is generally more visible further away from the coast, as shown in the upwelling case of Barrier et al. (2016) in the same area of study. There is no obvious link between wind intensity and the duration or the end of the event. However on the other hand, the wind intensity appears to drive the beginning of the upwelling as intense W appear when wind intensity equals or exceeds  $15\ m\ s^{-1}$ , which is a different threshold than cited in (Berta et al., 2018) ( $10\ m\ s^{-1}$ ). For both upwellings, their appearance is concomitant with the strongest SST drops on a two months time window (Zveryaev (2015) exhibit large intraseasonal SST fluctuations). SST observations a week before  $U_{2012}$  also exhibit the same drop, matching north-westerly wind and intense positive and negative W shifts lasting 12 hours, but not allowing an unambiguous upwelling characterization. SST satellite observations for  $U_{2022}$  were also compared to in situ surface temperature observations, highlighting negative shifts of  $\approx 1^{\circ}\mathrm{C}$  and  $\approx 2^{\circ}\mathrm{C}$  for satellite, and in situ observations respectively. These corresponded to the biggest temperature drops in two months for both surface observations. Despite the proximity to the coast, satellite observations appear consistent and are enhanced by SST in situ observations (Fig. 9).

Both downwellings appear to happen during a SLA maximum or a post maximum descent (all maxima lasting several days). With a shorter time window (12 hours) than upwellings (48 hours), both downwellings have averaged time-depth intensities higher than both upwellings while extreme values remain smaller than upwelling ones when no time average is applied. These time windows are purposely centered on the most intense W values of each studied event. We observe much more W variability through time in upwellings than downwellings. In opposition with upwellings, they also do not always cover the entire water column. The strongest negative W are rather concentrated between 20 and 60 meters depth in our case. Easterly and south-easterly wind episodes are less intense than north-westerly ones. Nevertheless, the downwellings highlighted here are triggered by wind intensity values around  $15 m s^{-1}$  such as for upwellings.  $D_{2014}$  matches intense positive meridional component of current, hence onshore horizontal currents, while  $D_{2021}$  shows a strong variability in direction for horizontal currents within two days. Twenty-four hours before  $D_{2021}$ , easterly wind matches a weak onshore current before shifting direction during  $D_{2021}$ , weakly heading offshore. A time lag is observed in  $D_{2021}$  between onshore currents and triggered negative W.

## 4 Discussion

On the one hand, the occurrences and intensity of biology-induced intense W highlight the importance of their filtering to obtain physical vertical velocities. On the other hand, studies with different objectives could on the contrary focus on these events in order to study them with the appropriate associated biological measurements.

It is the case in Guerra et al. (2019), where zooplankton diel vertical migrations (DVMs) were identified by matching mean volume backscatter strength and strong vertical velocities in upper layers, combined with zooplankton net samples. However, our biological signal differs from DVMs as the latter includes two phases: an ascent and a descent, associated with respectively positive and negative vertical velocities. At JULIO, we do not observe DVMs but only the negative W patches in a constant depth-layer between dusk and dawn. Some hypotheses are that these patches could be the sign of a sinking phenomena due to satiation of living scatterers that occurs during the night as mentioned in Tarling and Thorpe (2017) or that the angle of displacement of living scatterers could impact the returning signal of the ADCP (pers. comm. M. Ohman). The two days of upwelling ( $U_{2022}$ ) affect the usual pattern of biological signal at night (Fig. 8, right W) while the case of  $U_{2012}$  is less clear.

On Figure 7, as opposed to Figure 3, the months sampled can be identified and henceforth how they were impacted by biology-induced vertical velocities. As probability density functions (PDFs) highlight a faint positive skewness in vertical velocities measured in 2013 only, the depth-time average shows that this year was sampled only from late September to the end of December, with small influence of biology-induced negative vertical velocities. Moreover, a look at wind directions and intensities during this period (data not shown) indicates that: from September to December 2013, 49.5% of wind directions were north-westerly winds, and that 95% of the wind peaks (above  $10 \ m\ s^{-1}$ ) were also north-westerly winds. Overall, no interannual variability seems to emerge. The two full-data years, namely 2021 and 2022, show very similar distributions, featuring a majority of negative vertical velocities with a negative median. The two first criteria of our event detection method (intense W and consistent U and V) led us to discover a strong link between intense W and the presence of near inertial oscillations (horizontal currents direction shifts) which are almost ubiquitous in our time series. These features constitute an important part of intense W noticeable in Figure 7.

The described upwellings (48 hours) show averaged W with lower intensity values of  $1.1 \times 10^{-3}$  ( $U_{2012}$ ) and  $1.5 \times 10^{-3}$  m  $s^{-1}$  ( $U_{2022}$ ) compared to the downwellings (12 hours) with averaged W of  $-5.4 \times 10^{-3}$  ( $D_{2014}$ ), and  $-4.0 \times 10^{-3}$  m  $s^{-1}$  ( $D_{2021}$ ). This is an effect of the W variability driving the fact that the larger the observations time duration the smaller the averaged values. In particular, upwellings present a large time-variability with the presence of both positive and negative velocities covering the entire water column, lowering their averaged time-depth values.

As the first measurement of vertical velocities associated with upwelling or downwellings detected with an ADCP in the Northwestern Mediterranean Sea were presented here, it is noteworthy to underline different upwelling systems, and their associated vertical velocities, which have been studied in other coastal environments. Estimated W from  $\omega$ -equation allowed Ngo and Hsin (2024) in the Vietnamese upwelling system to reach values of the order of  $10^{-1}$  m  $day^{-1}$ . On the other hand, Johnson (1977) used the continuity equation to estimate W around  $10^{-4}$  m  $s^{-1}$  in the Oregon upwelling system. Such values are weaker than our observations. However, Mauzole et al. (2020) obtained vertical velocities up to 20 m  $day^{-1}$  in a coastal

upwelling in the California Current System area using a general circulation model. While earlier in the same area, Jacox et al. (2018) undertook a thorough review upwellings identification criteria, adding up Ekman and cross-shore geostrophic transports. In our area of study, Bakun and Agostini (2001) estimated vertical velocities greater than  $0.5 \ m\ day^{-1}$  at the bottom of the surface Ekman layer using the continuity equation. Other upwelling cases have been studied by Berta et al. (2018), linking their occurrences to strong (above  $10\ m\ s^{-1}$ ) north-westerly wind episodes lasting 2 to 3 days which is consistent with our model outputs (Fig. 8). Using temperature and wind observations, Odic et al. (2022) observed that upwellings and downwellings exhibit seasonality as they are less frequent during autumn and summer than during spring and winter which is consistent with our four observed events. The duration of both upwellings in this study (Fig. 8) is similar to the ones detected in Mourre et al. (2022), with a three-day-long upwelling (based on its effect on SST) or durations from 21 to 51 hours (based on a backscatter volume threshold). In terms of intensities, the latter exhibits episodic vertical velocity values between 60 and 80  $m\ day^{-1}$ , which is equivalent to about  $0.7 \times 10^{-3}$  to  $0.9 \times 10^{-3}\ m\ s^{-1}$ , about half the intensity of the upwellings shown here.

Moreover, these wind episodes are notably influenced by orography, meaning that easterlies do not impact the Gulf of Lion continental shelf and offshore the same way while north-westerlies seem more homogeneous, covering the entire Gulf of Lion (see Supplementary Materials). Easterlies can as well be confined off the continental shelf with north and north-easterly winds acting as a limit and thus creating an atmospherical front. Thus, the wind measured above JULIO point may not influence directly the currents observed at JULIO, but stronger winds at higher latitudes could.

Three independent methods have been used on June 24th 2022 to measure W at JULIO (Fig. 2). After comparison, measure-ments show W values with intensities smaller than during the above studied dynamical events, leading to the conclusion that no upwellings or downwellings were happening on that day. Nonetheless, we observe, with the FF-ADCP, that W could vary from  $-10^{-2} m s^{-1}$  to  $10^{-2} m s^{-1}$  in approximately 9 min at 20 meters depth (Fig. 2), highlighting a high variability of W on this day, probably caused by rapid internal waves. The use of a meteorological model as a prediction tool could help to set up dedicated cruises with the VVP and FF-ADCP during or right after a strong wind episode (such as Mistral or south-easterlies). Combining VVP and FF-ADCP methods would provide W data to be compared with JULIO mooring measurements as in section 3.1, on identified high-intensity events.

## 5 Conclusions

Vertical velocities (W) are measured with the JULIO moored-ADCP over the period 2012-2015 and since 2020. After filtering out intense biology-induced negative W events, the remaining high-intensity W events are analyzed, with a special focus on upwellings and downwellings. A dozen of these events (9 upwellings and 3 downwellings) were detected with an identification method. Two upwellings and two downwellings are thus thoroughly described, and show consistency between W and all the other available observations (wind model, satellite and in situ data).

This work shows that the vertical velocities can be measured with a moored-ADCP, and can be interpreted as fine-scale physical processes such as episodic coastal upwellings and downwellings in a non-rectilinear coastal site. Upwellings appear to be more easily characterized than downwellings, as they last longer, cover the entire water column and occur systematically

during intense north and north-westerly winds. Downwellings occur during easterly winds and mainly in the middle of the water column, only covering  $\approx$ 50 to 70% of it. Nonetheless in both upwelling and downwelling cases, regardless of the direction, wind intensities are likely to be higher or on the order of 15 m  $s^{-1}$ .

W measured with JULIO exhibited strong intensity values for both upwellings and downwellings. The instantaneous maximum amplitude measured is  $-1.93 \times 10^{-2}~m~s^{-1}$  for downwellings and  $2.55 \times 10^{-2}~m~s^{-1}$  for upwellings. A depth-time average covering each whole event shows maximum upward W as  $1.6 \times 10^{-3}~m~s^{-1}$  (48 hours averaged upwelling) leading to  $\approx 138~m~day^{-1}$  and maximum amplitude for downward W as  $-5.4 \times 10^{-3}~m~s^{-1}$  (12 hours averaged downwelling) which is equivalent to  $\approx -465~m~day^{-1}$ . The negative median values of years 2021 and 2022 underline the potential role of biology-induced vertical velocities as these years are the only two to have been fully sampled and arise the question of 2012 where biology-induced W seem to be scarcer.

Moreover, for the first time to our knowledge, two additional in situ W measurements were performed concomitantly and their analysis shows consistency between the three types of vertical velocity measurements. One of these three methods is about to be enhanced, as a new-generation ADCP (Sentinel-V RDI 500 kHz) with a fifth vertical acoustic beam directly measuring the vertical velocities has replaced the classical RDI OS ADCP 4-beam 300 kHz at JULIO early 2025, and should provide W measurements with an increased precision.

The filtering algorithm of biological-induced vertical velocities can be adjusted to DVMs, or to any region of the world ocean, coastal as offshore. Then the study of the remaining physics-driven vertical velocities will enhance our understanding of fine-scale oceanic processes, which have real consequences on how nutrients and other biogeochemical tracers get transported throughout the ocean.

Author contributions. MA: writing, data analysis, figures, data cleaning (JULIO 6th time series). AP: leader of JULIO observation program, supervision, writing, data cleaning (JULIO 1st to 5th time series). JLF: scientific cruises, mooring maintenance, review, suggestions. CC: data cleaning (FF-ADCP). AB: review, suggestions. YO: provided model output data, review, suggestions. SB: supervision, writing, data cleaning (FF-ADCP & VVP). All authors contributed to the writing of the original manuscript.

Competing interests. The contact author has declared that none of the authors has any competing interests.

Acknowledgements. We address a special thank to the MIO Physical, Littoral and Coastal Oceanography team (OPLC) for funding this work.

We thank Vincent REY, Tathy MISSAMOU and Didier MALLARINO, responsible of the HTMNet (data collection and processing). We thank Anthony Bosse through the GLISS project for funding the presentation of this work at an international conference. We acknowledge the MOOSE program (Mediterranean Ocean Observing System for the Environment) coordinated by CNRS-INSU and the Research Infrastructure ILICO (CNRS-IFREMER) for the support of the JULIO time series. We thank the Doctoral School of Environmental Sciences (ED251) for funding this work through the first author's PhD scholarship. We thank the captain and crew of the Antedon research vessel. We

| 470 | also thank Gilles Rougier for his precious help and work on JULIO's first time series. We address a special thank to Julie Gatti who, with her |  |  |  |  |  |
|-----|------------------------------------------------------------------------------------------------------------------------------------------------|--|--|--|--|--|
|     | pioneering PhD work, paved the way for such a study to be conducted.                                                                           |  |  |  |  |  |
|     |                                                                                                                                                |  |  |  |  |  |
|     |                                                                                                                                                |  |  |  |  |  |
|     |                                                                                                                                                |  |  |  |  |  |
|     |                                                                                                                                                |  |  |  |  |  |
|     |                                                                                                                                                |  |  |  |  |  |
|     |                                                                                                                                                |  |  |  |  |  |
|     |                                                                                                                                                |  |  |  |  |  |
|     |                                                                                                                                                |  |  |  |  |  |
|     |                                                                                                                                                |  |  |  |  |  |
|     |                                                                                                                                                |  |  |  |  |  |
|     |                                                                                                                                                |  |  |  |  |  |
|     |                                                                                                                                                |  |  |  |  |  |
|     |                                                                                                                                                |  |  |  |  |  |
|     |                                                                                                                                                |  |  |  |  |  |
|     |                                                                                                                                                |  |  |  |  |  |
|     |                                                                                                                                                |  |  |  |  |  |
|     |                                                                                                                                                |  |  |  |  |  |
|     |                                                                                                                                                |  |  |  |  |  |
|     |                                                                                                                                                |  |  |  |  |  |
|     |                                                                                                                                                |  |  |  |  |  |
|     |                                                                                                                                                |  |  |  |  |  |
|     |                                                                                                                                                |  |  |  |  |  |
|     |                                                                                                                                                |  |  |  |  |  |
|     |                                                                                                                                                |  |  |  |  |  |
|     |                                                                                                                                                |  |  |  |  |  |
|     |                                                                                                                                                |  |  |  |  |  |
|     |                                                                                                                                                |  |  |  |  |  |
|     |                                                                                                                                                |  |  |  |  |  |
|     |                                                                                                                                                |  |  |  |  |  |
|     |                                                                                                                                                |  |  |  |  |  |
|     |                                                                                                                                                |  |  |  |  |  |
|     |                                                                                                                                                |  |  |  |  |  |
|     |                                                                                                                                                |  |  |  |  |  |
|     |                                                                                                                                                |  |  |  |  |  |
|     | 22                                                                                                                                             |  |  |  |  |  |
|     | 23                                                                                                                                             |  |  |  |  |  |

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
