# Peer review of "Estimating oceanic physics-driven vertical velocities in a wind-influenced coastal environment"

_EGUsphere, 2025_

## Author Comment (AC1)

**Estimating oceanic physics-driven vertical velocities in a wind-influenced coastal environment**

Maxime Arnaud, Anne Petrenko, Jean-Luc Fuda, Caroline Comby, Anthony Bosse, Yann Ourmières, and Stéphanie Barrillon

**Reply to referee #1: Diego Cortés-Morales**

We would first like to thank you for the time you spent reading our manuscript and your constructive comments. Before answering, please note that we decided to add the latest JULIO time series (from July 12, 2023 to May 22, 2024) to our revised manuscript as we obtained the data during the review process. We think that it brings valuable information on our work as it adds ~20% of data compared to the previous dataset in the original version of the paper.
 Following your suggestions, we have modified our manuscript and answered your comments.

Specific comments:

The manuscript repeatedly states that JULIO has measured vertical velocities "since 2012". This phrasing is misleading because the dataset is discontinuous, with a gap between 2015 and 2020 (see Table 1 and Figure 10). This limitation should be explicitly acknowledged in the abstract and the rest of the manuscript, and do not consider the analysis of more than a decade of data.
You are absolutely right: we made it more obvious in different parts of the text.
New lines 6-7: " JULIO provides Eulerian measurements of three-dimensional current velocities over two time-periods: 2012-2015, and since 2020."
New lines 123-124: "The JULIO (JUdicious Location for Intrusion Observation) mooring ADCP provides in situ time"
New lines 151-152: "The observations (interrupted by yearly maintenance and trawling incidents) began in 2012 and were suspended from 2015 to 2020."
New line 433: "Vertical velocities (W) are measured with the JULIO moored-ADCP over the period 2012-2015 and since 2020."

The comparison between JULIO ADCP, FF-ADCP and VVP is presented as a central result, but the conclusions are too general and not the core objective of this paper. The conditions for the comparison should be highlighted and clarified, including the choice of periods used for JULIO ADCP and FF-ADCP of around 2 hours while VVP data is limited to 17 minutes. Could more detailed insights be extracted from this intercomparison? For example, how each method captures upward vs. downward velocities, or how the sign of *w* evolves in time?
The way the results were presented was indeed unclear and was changed thanks to your suggestions. The main result from this comparison is that the order of magnitude and variability are consistent between these measurements. Hence no bias is seen for each measurement compared to the others. Nonetheless, the different methods capture different time properties. We have added a figure (Fig. 2 bottom) to illustrate this point.
*In situ* measurements are highly dependent on meteorological conditions, as well as other time or strategy constraints. In our case, only one cast of VVP was possible because of time constraints of its deployment: 15min at the surface before diving (to be able to have the GPS signal), then diving at 80 meters at +/- 10 cm/s lasting about 15 mn and back up (measurement phase) during ~17 minutes. Hence the measurements of JULIO ADCP (every ½ hour) are chosen to be the closest available to this single VVP cast.  As for the FF-ADCP casts, 1 hour of mission was planned during the VVP cast.

The manuscript does not discuss the intrinsic measurement errors of the instrument. What are the uncertainties associated with the velocity estimates (e.g. Figure 3)? Given the small magnitude of the measured vertical velocities in the intercomparison, this information is essential.

The error measurement of vertical velocities combines the instrumental and the methodological errors. In our case, the errors are the following:

- Intrinsic ADCP error due to hardware which includes (for our ADCP 300 kHz) 0.5% of the measured value and a quadratic addition of a conservative value of +/- 5 mm/s, both given by the manufacturer.

- Methodological ADCP, depending on the deployment characteristics: number of bins (cell size), number of pings. This conservative error (private communication with RDI) is +/- 5 mm/s, added quadratically

The combined error gives us the absolute instrument error of +/- 7mm/s.

The errors coming from the biological identification can also influence the average W calculated in each specific event. For instance, we computed W averaged over $U_{2022}$ for both smaller and bigger boxes, giving a difference of 2.8 mm/s between these two extremes (new lines: 314-316).

The manuscript is somewhat disorganised. Very specific event analyses are introduced early, without giving a proper explanation of this choice, while a broader overview of the JULIO time series only appears near the end (Figure 10). A more logical structure would be to first present the general time series (Figure 10 or even Figure S18) after Figure 3 and then focus on the specific events.
Indeed the manuscript lacked fluidity. In the new version, after the comparison between the three methods to place a context around the values obtained at JULIO, we give an inter annual overview of all vertical velocities combined (physics-driven and biological-induced) to show the phenomena that can be observed there. Then we explain and apply our biology filtering method, showing a specific example before applying it to all time series. Then we present our identification method for upwelling/downwelling events and focus on four specific examples.

Moreover, as you recommended, we chose to keep only the depth averaged version of Fig. 10 (now Fig. 7), and not the time-depth averaged of the initial version, hence presenting more general time series.

The annual probability distributions in Figure 3 should be interpreted with caution. According with Table 1 and Figure 10, only 2021 and 2022 provide complete annual coverage, so interannual comparison are likely dominated by sampling differences and seasonality of the vertical flow. The statement on lines 234 and 235: "boxplots highlight that 50% of the values are between -5 and 5x10^(-3) m s-1" should be framed as an order of magnitude, not a strict threshold. The manuscript should also clarify that the variability in the number of observations is due to short time series than a different sampling frequency (this is mentioned only later, on Line 316 but should be brought to the reader much earlier).
The sampling context was not explicitly reminded to the reader when the distributions were analyzed. We changed our descriptions/statements accordingly:
Line 252-253: "Annual sampling varies from one year to another regarding the duration, the number of months, and the sampled seasons."
Line 256-257: "Nonetheless, one should note that, although the sampling frequency remains the same, the number of months sampled varies depending on the year."

The criteria for selecting the upwelling and downwelling events, as well as the period used for the identification (Line 302) are nor well justified. The chosen events do not appear to be the most intense in Figure 10. For example, in U2022, why do the authors chose two days for the consideration of the upwelling? Do the authors use the wind velocity to identify these events? If so, how sensitive are the results to these choices? For U2012, the extreme wind event above 15 m s-1 is shorter than for 2022

and lasts less than 2 days. I don't understand the choosing of the window because it starts after a 15 m s-1 measurement, but it is not contained in the considered period. Without a statistical analysis of the entire time series, the representativeness of these four events is questionable, and the conclusions should not be general.

As mentioned in lines 415 to 417, different criteria can be used to determine an upwelling or downwelling duration. It is true that our choice was not clearly explained as well as our identification method (now detailed from line 298 to 305). For the example of U2012, we chose a window that matched intense wind (above 10 m/s as a threshold) and the W response (intense and positive values). For each example presented in the paper, we focused rather on the duration of intense W than the duration of the wind gust: for example, D2014 was triggered by a wind gust that did not last the entire downwelling.

The average upwelling and downwelling values reported in the abstract and the conclusions are extremely sensitive to the chosen filter parameters and averaging window. Moreover, they are based on a very small subset of events (one single event of 2 days and 12 hours respectively) relative to 5-year dataset.

That is true. We decided to keep a no-time averaged version of Fig. 10 (now Fig. 7) to exhibit the occurrences of strong W and our studied events. We also precise in the text that 12 events have been detected with our detailed identification method and four of them have been chosen as examples (new lines 306 & 435). We chose the four events detailed in the paper as illustrative examples for their good agreement with satellite observations.

Technical corrections:
The manuscript requires careful proofreading, especially regarding comma usage. Below are some specific corrections and suggestions:
All these corrections and suggestions have been taken into account.
• Line 11: "associated with" (not "associated to")
New line: 11
• Line 12: Use consistent notation for vertical velocity, here you use lower-case w, but during the rest of the draft you refer as W.
Deleted.
• Line 12: "Hence," (add comma).
New line: 12
• Line13: "high-frequency"
Deleted.
• Lines 34 and 42: "of a few" (not "of few")
New lines: 35 & 44.
• Line 35: "allow estimating" instead of "allow to estimate"
New line: 37.
• Line 38: "Frajka-Williams et al. (2011) were"
New line: 40.
• Line 39: "Other 3-month" (not "Other 3-months")
New line: 41.
• Line 44: change "since decades" to "for decades"
New line: 45.
• Line 50: change "electro-magnetic" for "electromagnetic"
New line: 52.
• Line 50: Clarify whether APEX is an acronym?
Clarified line 52: "Using electromagnetic APEX (Autonomous Profiling EXplorer) floats"
• Line 61: change "ship mounted" for "ship-mounted"
New line: 63.

- Line 64: change "inter annual" for "interannual"

New line 66.

- Line 84: "downwelling and upwelling events" (not "downwellings and upwellings events" )

New line 85.

- Lines 94 and 196: Could you  maintain the same format for coordinates?

We made sure to re-write every coordinates in the same order, and in decimal degrees.

- Line 97: change from "vertical velocities measurements" to "vertical velocity measurements"

Changed.

- Lines 100, 119, 270: "W was…, W is …, W was" W is referred as plural in the rest of the manuscript

Line 100 (new line 101): corrected to "were"

Line 119 (new line 126): "W is" the "is" is not for W but for "analysis". The whole sentence is "Within the framework of these observations, a thorough analysis of the intensity and variability of W is conducted using the moored ADCP time series dataset."

Line 270 (new line 310): "W was" refers to an upwelling here. The whole sentence here is "An upwelling (U2022) with intense positive W was detected at JULIO, after biology filtering, from April 1st to April 3rd 2022"

- Line 102: Add comma "Generally, W have very low intensities"

Changed.

- Line 112: change from "also impact" to "also impacts"

New line: 111.

- Line 116: Introduce the JULIO acronym at the first mention, not in line 135

Introduced in the abstract (line 5) and intro (line 123).

- Line 117: change "multiyears" to "multi-year"

Deleted.

- Line 117, title Table 1, 191 and 409: "time series" (not "time serie")

New lines: 123, 202 and 395.

- Lines 122 – 24: No space before a question mark in English.

New lines 128-130.

- Line 127: Introduce the FF-ADCP acronym at the first mention

New line: 132.

- Line 144: Which is the sampling frequency of JULIO? I think it is written in line 230, can you put it here instead?

The sampling frequency of JULIO was explicited at line 143 (now 149) "Time series are obtained with a time resolution of half an hour and a vertical resolution (i.e. cell size) from 4 m to 2.5 m (Table 1)." We think it is important to keep it in the inter-comparison section too, as we insist on the differences of sampling between the three methods (new line 247).

- Line 150-157: Missing reference

New line: 162-163.

- Line 160: Justify parameter choices with references.

We chose to apply a rolling mean to obtain the EAA trend as the signal was too noisy to be exploitable. We didn't want to take a larger window to avoid seasonal variability. The 15-day rolling mean was the optimized parameter for our study, sufficient to avoid noise and not too large to avoid seasonal changes.

- Line 164: Define "MIO".

New line: 173-174.

- Lines 188 to 190: Missing commas before numbers?

New lines: 198-201.

- Figure 2: Could the authors thicken the dashed lines in JULIO and FF-ADCP panels? In the FF-ADCP case, I think that thickening the lines would improve the readability of the different colours. Clarify meaning of lines in caption (depth units, time periods, shading). Could you add also the time period of each methodology in the caption, will be easier for the reader than to search for it in the body text.

Changed for Fig. 2.

- Line 245: Remove double parenthesis in reference: "((Heywood, 1996))".

New line: 268.

- Line 247: Missing space in "(Fig.6 top panel)".

New line: 270.

- Figure 4 caption: There is not solid lines in the figure, do you mean dotted lines?

Indeed, we forgot to change our descriptions. The Fig. 4 caption has been changed.

- Line 252: remove space "value :"

Deleted.

- Figure 6 caption: Could you add the period of study as in Figure 4?

The caption has been changed.

- Line 260: Replace "penultimate panel" with "panel c)".

New line 283.

- Line 264: Could you add a reference? This affirmation is for the climatology. Is always constant during the year or it has a seasonal cycle?

Thanks for underlining it. It indeed has a seasonal cycle. The appropriate bibliography has been added (new line: 295-296).

- Line 266: Reference needed for the 15 m s-1 threshold

The wind threshold and its reference has been detailed in our upwelling/downwelling identification method (new line: 303).

- Line 275: Use American spelling "analyzed" consistently

New line: 320.

- Line 277: Add space in "(Fig.8)".

New line: 322.

- Line 284: I do not think that it is necessary to add <W_2012 >, if you do, please add it in the other cases <W_2022 >.

Indeed, we decided not to use this writing (new line 329).

- Figure 7. Improve colour contrast of the lines U2012 and U2022. Could you add the years 2012 and 2022 above each column as title? It would help with reading. Could you add letters to the panels in the figure to make it easier to refer to them in the text? Adding a horizontal line at 15 m s-1 wind panels could help to define better the average region.

- Figures 7 and 9. The authors are using a sequential colormap for a variable with positive and negative values. I recommend using a diverging colormap to improve readability and the understanding of the discussion and conclusions.

The Figures 7 and 9 (now 8 and 10) have been changed accordingly.

- Figure 10 caption: Clarify whether horizontal lines correspond to +- 5 mm s-1?

Figure 10 (now Fig. 7) caption has been changed.

- Figure 10: Ensure consistency between Figure 10 and Table 1 (dates for 2013–2014). Time series during 2013 and 2014 do not correspond with the dates of Table 1 (Time series 2). In the table is written 03-26-2013 as initial date, but the time series in Figure 10 show that the values start at the end of September 2013.

Thank you very much for pointing this. We corrected it.

- Line 321: Add missing space and remove dot: 15m.s−1

New line: 391.

- Line 324: Add missing units: 0 m s−1

New line: 296.

- Line 325: Add missing space: 5 x 10^(-3m s−1

New line: 288.
- Line 327: remove s in "exhibits"

New line: 344.
- Line 343: Consider citing Jacox et al. (2018) on coastal upwelling.

Thank you for this recommendation. Jacox is cited in new lines 407-408.
- Line 346: Missing r in "occurences"

New line: 412.
- Line 360: Justify choice of a two-month time series? Consider seasonal climatology

We centered the satellite time series on our events. The chosen window had to be large enough to observe variations or trends but not too large to avoid seasonal variations. Thus, we chose a two-month window for clarity.

- Line 371: Meridional component during D2014 event seems much more variable than for D2121 event. Maybe it is because of the colormap chosen does not allow for a clear view of positive and negative values as I mentioned above.

Thank you for the suggestion, indeed a diverging colormap allows us to better distinguish currents direction variations. Regarding the variability, D2014 matches intense positive meridional component and thus onshore current. The variability here seems to be more in intensity than in directions (except the two deepest layers). For D2021, not only the meridional component but the horizontal currents are more variable in directions and not in intensity: zonal component shifts direction during D2021 and meridional component shifts at the very beginning of the event.

We cleared the text in that way: new lines 369-370 "D2014 matches intense positive meridional component of current, hence onshore horizontal currents, while D2021 shows a strong variability in direction for horizontal currents within two days."
- Line 391: Remove s in "works"

New line: 438.
- Lines 390-396: The authors should detail that they are talking about the four events analysed and only generalised to the entire period of study after demonstrating that they are representative events. Could you compute the correlation coefficient between wind velocity and vertical velocity?

A Pearson correlation has been performed between wind and intense W but we found a weak correlation, which is not surprising giving the fact that intense W exist without being related to strong wind events and upwelling or downwelling events.
- Line 401: remove s in "medians"

New line: 448.
- Line 405: missing s in "type"

New line: 452.

---

## Author Comment (AC2)

**Estimating oceanic physics-driven vertical velocities in a wind-influenced coastal environment**

Maxime Arnaud, Anne Petrenko, Jean-Luc Fuda, Caroline Comby, Anthony Bosse, Yann Ourmières, and Stéphanie Barrillon

**Reply to referee #2**

We are sincerely grateful to you for the time you spent reviewing our work and for your helpful suggestions. Before answering, please note that we decided to add the latest JULIO time series (from July 12, 2023 to May 22, 2024) to our revised manuscript as we obtained the data during the review process. We think that it brings valuable information on our work as it adds ~ 20% of data compared to the previous dataset in the original version of the paper.
We changed our manuscript accordingly.

**Specific comments**
1.      Line 224: avoid using the root word "significant" in scientific writing if no statistical tests are conducted. Appropriate statistical tests should be conducted before concluding that the results of all measurement methods are not significantly different from one another. Additionally, the use of scientific notation is not needed if the value is zero (refer to $0.0 \times 10^{-3}$).
We agree, the word significant has been removed (new line 234).
We also changed the scientific notation for zero (new line 234).

2.      A sensitivity analysis of the wind speed threshold helps in validating the $15$ m s$^{-1}$ choice. This is relevant since as mentioned in line 346, Berta et al. (2018) have shown that wind speeds above $10$ m s$^{-1}$ are strong enough to drive upwelling in the authors' area of study.
You are right we have decided to follow the wind threshold of 10m/s from Berta et al; We explained our event selection (from line 298 to 305) as follows:
"This work focuses on upwelling and downwelling events. Once the biological-induced signal is filtered, physics-driven events identification follow four criteria in the following order. We first identify groups of physics-driven vertical velocities with intensities greater than ± 10 mm/s lasting at least 2 hours (5 values) and spreading deeper than ≈ 10 m. If the criterion is met, we look for groups of horizontal velocities in the proximity of the identified W, that match both intensities greater than ± 200 mm/s and offshore (onshore) directions, consistent with an upwelling (downwelling) event. If these two criteria are met, we analyze wind speed intensity and direction from ARPEGE model, keeping episodes where intensities are over 10 m/s (Berta et al. 2018) and directions are north-westerly (south-easterly) to match offshore (onshore) horizontal
currents. Finally, this whole identification is completed by an analysis to detect the expected variability in SST (SLA) satellite observations leading to the upwelling (downwelling) event identifications."

Nonetheless we also observe that our chosen events corresponded to stronger gales (15 m/s).

3.      Lines 11–13: the authors in the abstract highlight how the order of magnitude of W depends on the spatiotemporal scale of its analysis and also the need for high-frequency measurements. The statements are not only unclear, but also feel random and out of place because there is hardly any discussion on how different sampling frequencies affect the final results of the study.

Thank you, we have deleted the statement in the abstract as our Fig 10 (now Fig. 7) is not temporally averaged over a 4h time window anymore.

The frequency measurements at JULIO (48 per day) is adequate for the physical processes (upwelling

and downwelling events) targeted in this paper. FF-ADCP measurements, with a sampling every 2min, would have been a more appropriate tool if we had chosen to study internal waves for example.

4.      Section 3.6 and Figure 10 should be placed before Section 3.5 and Figures 7–9 for better structure and flow. Describe the general trend and features (e.g. the total number of upwelling and downwelling events identified in the entire time series) before moving on to describe the specific events in detail.

We have followed your suggestion. The manuscript's flow should be more fluid now. In the new version, after the comparison between the three methods to place a context around the values obtained at JULIO, we give an inter annual overview of all vertical velocities combined (physics-driven and biological-induced) to show the phenomena that can be observed there. Then we explain and apply our biology filtering method, showing a specific example before applying it to all time series. Then we present our identification method for upwelling/downwelling events and focus on four specific examples.

5.      I also suggest adding the adjective "physics-driven" in the title to better reflect the scope of your study.

Thank you, the new title definitively reduces ambiguities.

6.      The organisation of the manuscript can be further polished and requires a thorough proofreading. There are many short paragraphs that are 1–2 sentences long (e.g. lines 54, 93, 339, 350, 387, 389, 404) that could instead be integrated with other relevant text to produce longer coherent paragraphs.

Indeed, we had split paragraphs into separate ones without it being necessary. The new manuscript should be more readable now.

7.      Some descriptions and ideas are repeated often, thereby reducing the brevity of the manuscript. One notable example is the Discussion section, where some paragraphs are largely a rehash of the results presented in section 3.5. To improve the structure of the paper and make it more concise, here are some key suggestions:
•       Move lines 96–99 to section 2.2.
•       Move section 3.6 in front of section 3.5 (as mentioned earlier).
•       Move the bulk of the description in lines 324–333, lines 354–363, and lines 365–374 to section 3.5, and reduce the repetition.
•       Lines 337–338: sentence is phrased like a summary statement that is placed awkwardly in the middle of the Discussion section. I suggest removing it.

Thank you for your suggestions.

- We decided not to move lines 96-99 to section 2.2 as we wanted to describe the rather scarce W estimations in our area of study in the introduction.
- The section 3.6 has been moved before section 3.5 to improve fluidity of the article.
- The description lines (old version; line 324–333, lines 354–363, and lines 365–374) have been moved to a new section (3.7 new lines 343-348, new lines 349-357, new lines 362-372) prior to the discussion section, which includes the comparison of the four events characteristics.

- The sentence (old version; lines 337–338) has been rephrased to "As the first measurement of vertical velocities associated with upwelling or downwellings detected with an ADCP in the Northwestern Mediterranean Sea were presented here, it is noteworthy to underline different

upwelling systems, and their associated vertical velocities, which have been studied in other coastal environments" (lines: 402-404), as a transition to introduce other upwelling systems in different areas.

**Technical corrections**
Thank you, all these comments have been taken into account according to your suggestions.
Line 1: change to "oceanic vertical velocities (W)".
Changed.
Lines 11–12: Write "w" in caps.
Deleted sentence.
Line 13: remove "spatiotemporal".
Deleted sentence.
Line 18: change to "allowed for".
New line: 20.
Line 22: awkward use of colon.
New line: 24.
Line 24: the word "very" is redundant since "complete" is already an absolute adjective.
True. The sentence has been rephrased to "A complete summary of submesoscale mechanisms, […]", new line 26.
Line 27: change from "during the last decades" to "over the decades".
New line: 29.
Lines 35, 40, 364, 383: the phrase "allow to" is used awkwardly, please rephrase.
- Line 35, new line 37 changed to "allow estimating".
- Line 40, new line 42 changed to "With other 3-month observations in the same area by Clément et al. (2024) measured, using gliders, downwelling (upwelling) convective plumes [...]"
- Line 364, new line 361 rephrased to "Despite the proximity to the coast, satellite observations appear consistent and are enhanced by SST in situ observations (Fig. 9).".
- Line 383, new lines 428-429 rephrased to "Using a meteorological model as a prediction tool could help to set up dedicated cruises [...]"

Lines 41, 76, 88, 234, 326, 334, 367, 373: I suggest against starting a sentence with "indeed". It is a filler word that hardly adds or change any meaning to the overall flow.
Thank you for pointing this. We changed it accordingly:
- Line 41, new line 43-44 rephrased to ". In Christensen et al. (2024), Argo floats were used to estimate values […]"
- Line 76, new line 78 "Indeed" has been removed.
- Line 88, still line 88 "Indeed" has been removed.
- Line 234, new line 255 "Indeed" has been removed.
- Line 326, new line 344 "Indeed" has been removed.
- Line 334, new line 400 "Indeed" has been replaced by "In particular".
- Line 367, new line 366-367 the sentence has been rephrased. " We observe much more W variability through time in upwellings than downwellings."
- Line 373, new line 372 "Indeed" has been removed.

Line 66: awkward use of colon. The part after the colon is also repetitive and redundant.
New line 67 the colon has been removed.
Line 72: "…, seasonal variability affecting its width, …" – awkward phrasing.
New lines 73-74, rephrased to "Originating from the Ligurian Sea, this density current follows the coast with a horizontal speed from 0.4 up to 0.7 m/s in winter, with a width, depth and flow rate that are prone to seasonal variability (Petrenko, 2003)."
Line 121: no need for a new paragraph.
New line 127, changed.

Line 144: change to "information".

New line 149.

Line 149: remove the extra blank line.

Deleted, new lines 153-154.

Line 156: add a space between number and unit (i.e. 4 m).

New line 161.

Line 168: add a space between number and unit (i.e. 2 Hz).

New line 177.

Line 183: spell out 8 and write one as a numeral. For items other than units of time or measure, use words for cardinal numbers less than 10. Add a space between number and unit (i.e. 200 m).

Thank you for your comment. The changes have been applied in new line 193.

Here, we refer to attitude angle as used in Comby et al. 2022.

Line 195: change to "10 m".

New line 205.

Line 210: change to "resulting in".

New line 220.

Line 238: change to "example".

New line 259.

Line 240: present your results in prose instead of a bullet point list.

Changed from line 260 to 263. "Strong negative vertical velocities appear in patches (with averaged W in the patches = $-1.8 \times 10^{-2}$ m/s), describing a diurnal cyclespanning ≈8 hours and centered at midnight. Those patches are located mainly under the surface (between the surface and ≈50 meters depth) with a seasonal variability and a more pronounced presence in springtime (Fig. 5)."

Line 243: remove "a".

The sentence has been changed (see previous comment).

Line 267: remove "respectively" in parentheses.

The entire sentence has been removed.

Line 268: awkward use of the word "completed". Sentence is thus unclear.

New lines: 304-305. The sentence has been removed as we detail our event identification method instead.

Line 275: change from "of" to "in". Use "of" to describe the magnitude.

New line 320.

Line 285: change to "small decrease of".

New line 330.

Line 295: change to "24-day long".

New line 340.

Line 302: change to "variability of".

New line 348.

Line 331: change from "upwelling and downwelling W" to "upwelling and downwelling events".

The sentence has been removed.

Line 348: remove "a".

New line 414.

Line 365: change to "With a shorter".

New line 364.

Line 383: sentence is unclear with some awkward phrasing.

New lines 428-429. The sentence has been rephrased to "The use of a meteorological model as a prediction tool could help to set up dedicated cruises with the VVP and FF-ADCP during or right after a strong wind episode (such as Mistral or south-easterlies)."

Line 402: "arise the question of 2012" – unclear.

We noticed that biology-induced W during spring and summer in 2012 are less numerous than other years (2021 to 2023). If we compare the number of biology-induced W (number of green dots on Fig. 7 which was previously Fig. 10) from April to June for each of these years (same number of

measurements), we find:
- 255 in 2012
- 749 in 2021
- 806 in 2022
- 353 in 2023

For now, we don't have the necessary perspective to interpret these differences but long-term observations might be useful to conclude if 2012 is an exceptional year.

Line 403: write "w" in caps.

New line 450.

Line 404: change to "for the first time".

New line 451.

Lines 406–407: add a space between the number and kHz.

New lines 453-454.

Figure 1: intensity or velocity? Add a scale bar. Are the locations of the VVP and FF-ADCP the same as the JULIO mooring? If not, label them on the map.

The Figure 1 shows the intensity of horizontal velocities. A scale (10 km) bar has been added. The VVP and FF-ADCP locations are the same as the JULIO mooring; we precised it at line 230 to reduce ambiguities "Three W measurements have been made simultaneously in space and time at the JULIO site in 2022: JULIO's mooring ADCP, VVP, and FF-ADCP (Fig. 2)."

Figure 2: the font size of the title is too small.

We increased the titles font sizes.

Figure 6: in the caption, the panel labels (a) to (d) should precede the description of each figure panel, not after.

Thank you for pointing this. We changed the caption accordingly.

---

## Author Comment (AC3)

Estimating oceanic physics-driven vertical velocities in a wind-influenced coastal environment

Reply to community comment: B. Blanke

Maxime Arnaud, Anne Petrenko, Jean-Luc Fuda, Caroline Comby, Anthony Bosse, Yann Ourmières, and Stéphanie Barrillon

We would first like to thank you for your commitment and detailed comments about this paper. Your help has been sincerely appreciated. Before answering, please note that we decided to add the latest JULIO time series (from July 12, 2023 to May 22, 2024) to our revised manuscript as we obtained the data during the review process. We think that it brings valuable information on our work as it adds ~ 20% of data compared to the previous dataset in the original version of the paper. Following your suggestions, we have modified our manuscript and answered your comments. We detailed our manuscript modifications below.

1) The abstract should better highlight the key findings and their broader implications listed below:

(i) **Methodological work:** The authors combine three different measurement techniques (JULIO ADCP, FF-ADCP, and VVP) and introduce an innovative approach using echo intensity to separate biological noise from physical signals. This demonstrates that bottom-mounted ADCPs can provide reliable vertical velocity measurements, but only when biological activity in the water column is properly accounted for.

(ii) **Scientific results:** Measuring vertical velocities in the ocean is notoriously difficult, yet these motions are crucial for understanding ocean dynamics. The authors document vertical velocities for upwelling and downwelling events that transport water several hundred meters per day on average. This has real consequences for how nutrients and other biogeochemical tracers get transported throughout the ocean.

(iii) **Thorough validation:** Over a decade of data is cross-validated against satellite observations and model outputs. The study demonstrates that multiple types of measurements (acoustic, satellite, and meteorological) are necessary to understand the processes that occur in this coastal system.

We have changed the abstract accordingly.

"Despite the challenge of measuring them due to their small intensities, oceanic vertical velocities constitute an essential key in understanding ocean dynamics, ocean-atmosphere and biogeochemistry interactions. Coastal events and fine-scale processes (1-100 km / days to weeks) can lead to high-intensity vertical velocities.

Such processes can be observed in the Northwestern Mediterranean Sea. In particular, the Gulf of Lion is a region prone to intense north-westerly and easterly wind episodes that strongly impact the oceanic circulation. The JULIO mooring (JUdicious Location for Intrusion Observation) is located on the boundary of the Eastern side of the Gulf of Lion's shelf at the 100m isobath. JULIO provides Eulerian measurements of three-dimensional current velocities over two main time-periods: 2012-2015, and since 2020. Vertical velocities measurements from JULIO show a good agreement with two independent methods, a Free-Fall Acoustic Doppler Current Profiler and an innovative Vertical Velocity Profiler.

To measure physics-driven vertical velocities, we developed a method to identify and filter out biology-induced vertical velocities. Combining satellite and in situ observations with wind model outputs, we identify wind-induced downwelling and upwelling events at JULIO associated with physics-driven vertical velocities with maximum amplitudes of -465/127 m/day. Hence, this analysis underlines the need for long term multimethod observations in such coastal areas forced by intense wind episodes. This work presents mooring ADCPs as reliable tools for physics-driven W measurements, with an adaptive algorithm which is applicable anywhere offshore in the ocean to detect W in fine-scale processes."

**Issues:**

**Interpretation of the biological signal:** The study attributes recurring nighttime negative vertical velocities to biological processes but immediately dismisses diel vertical migration (DVM) because only descending motion is observed. Alternative biological explanations could be explored, such as asymmetric DVM behavior, sinking dead biomass, or ADCP backscatter bias due to the geometry of biological material. Could the authors present additional biological data (collected during patch events) to validate the assumption of a biological origin?

We were not clear in the original text. DVM consist generally in ascending velocities at dusk, and descending velocities at dawn. Here, at JULIO, we do not observe these classical DVM; but at certain periods of the year (generally spring & summer), during the night, we observe biological groupings or patches, which origin we ignore. The peculiarity of these groupings is that they appear rather stationary in space (subsurface layers), despite exhibiting overall negative velocities.

This kind of negative W patches have been observed and is not usually considered as DVM. We cleared the text in this way (new lines 114 to 120):

"Between the ascending (descending) phases at dusk (dawn) of DVMs, the zooplankton stays at the surface or subsurface during the night. These stationary patches are sometimes associated with recorded ADCP negative vertical velocities (Tarling et al 2001, BioSWOT-Med and FUMSECK cruises (unpublished work)). Some hypotheses attempt to explain these counterintuitive negative vertical velocities of stationary patches. Among them, a sinking phenomenon due to satiation of living scatterers that occurs during the night as mentioned in Tarling and Thorpe (2017) or the angle of displacement of living scatterers, varying depending on whether they ascend or descend in the water column thus impacting the ADCP returning signal (pers. comm. M. Ohman)."

We have also exchanged with biologists about observations which could be useful for upcoming work, including biological sampling at nights in the JULIO area.

**Methodological choices:** The 15 m depth × 4 h time window is based on an open-ocean study in the Scotia Sea. Should this choice be more carefully validated for coastal studies such as those in the Gulf of Lion? How might the bin size affect the detection of vertical velocity structures, particularly near the seabed or near the surface? Similarly, how do wind curl and coastal geometry influence Ekman transport at this specific site compared to open-ocean conditions?

The virtual window dimensions (12 m depth and 4 hours long) are inspired from Tarling 2009 but a sensitivity analysis has been made with our data: we varied depth from 8 to 20 m, time from 2 to 8 hours, and percentage from 50 to 90%. We kept the optimized parameters that are able to target our observed patches (which take into account the different bin sizes over the time series).

The algorithm is optimized for our specific study case and its characteristics would need to be changed for studies in open ocean or in other geographical zones. The method itself (sliding window to identify patches) is easily exportable (new lines: 281-285 & 312-317).

Ekman transport is indeed influenced by wind characteristics and localization. In our specific case, orography, with the Alps close to the sea, coast orientation, together with wind characteristics, added to the fact that JULIO is on the edge of the continental shelf thus influencing Ekman transport and the occurrences of upwellings and downwellings. Open-ocean conditions are very different, an ADCP mooring there would measure W from other physical processes.

**Measurement uncertainty:** The claimed minimum resolvable vertical velocity approaches many measured values, which requires rigorous quantification of standard deviations and biases for each method, particularly the JULIO ADCP. The study partially addresses this issue but lacks discussion of biologically active layers and systematic error propagation assessment. For instance, wind measurements contain uncertainties (instrument precision, space and time sampling limitations)… that can project onto vertical velocity estimates.

The error measurement of vertical velocities combines the instrumental and the methodological errors. In our case, the errors are the following:

- Intrinsic ADCP error due to hardware which includes (for our ADCP 300 kHz) 0.5% of the measured value and a quadratic addition of a conservative value of +/- 5 mm/s, both given by the manufacturer.

- Methodological ADCP, depending on the deployment characteristics: number of bins (cell size), number of pings. This conservative error (private communication with RDI) is +/- 5 mm/s, added quadratically

The combined error gives us the absolute instrument error of +/- 7mm/s.

The errors coming from the biological identification can also influence the average W calculated in each specific event. For instance, we computed W averaged over $U_{2022}$ for both smaller and bigger boxes, giving a difference of 2.8 mm/s between these two extremes (new lines: 314-316).

The wind model, on the other hand, is only used to identify upwelling and downwelling events.

**Statistical rigor:** The 15 m/s wind intensity threshold requires statistical or bibliographic justification. How sensitive are the results to this threshold? Can its validity be statistically tested? More generally, several conclusions rely on visual interpretation (wind-vertical velocity alignment, SST drops, SLA peaks) rather than on quantitative correlation or thorough analysis. Could the authors strengthen some of their interpretations for added robustness in their results?

Indeed our threshold was not clearly established. We use the strong wind intensity value from Berta et al 2018 (10 m/s) to select our events. 12 events are selected with all the parameters (new lines 298 to 305 in the text). Nonetheless we also observe that our chosen events correspond to stronger gales (15 m/s) before or during the upwelling/downwelling event. A Pearson correlation has been performed between wind and intense W but we found a weak correlation, which is not surprising giving the fact that intense W exist without being related to strong wind events and upwelling or downwelling events. Regarding SST and SLA, direct correlations with vertical velocities are tricky as satellite observations are daily and JULIO data features a sampling frequency of 48 times a day. This highlights the need, in our case, to systematically use multiple observations to identify such coastal events.

**Event selection and generalizability:** Were other upwelling/downwelling events identified by other methods and excluded from this analysis? If so, what criteria disqualified them? How generalizable are the documented characteristics (duration, depth extent, SST signatures) to other Mediterranean coastal systems? The conclusion could acknowledge that localized JULIO observations may not represent basin-scale processes and discuss the applicability of their method across the Gulf of Lion.

Thank you for your comment. We have now described precisely our method of detection of up- and downwellings (new lines 298 to 305). This way, we detected 9 upwellings and 3 downwellings.

We chose the four events detailed in the paper as illustrative examples for their good agreement with satellite observations. These events indeed do not represent basin-scale processes, but the conclusion is that this method (moored ADCP and biological filtering) could be used to measure physics-driven vertical velocities in open seas and assess other physical processes than upwellings and downwellings (for example internal waves, eddies, fronts or filaments).

We added information both in abstract and conclusion about the generalization of our observations to different basins.
New lines 13-15 "This work presents mooring ADCPs as reliable tools for W measurements when combined with other types of observations, with an adaptive algorithm which is applicable anywhere offshore in the ocean to detect W fine-scale processes."
New lines 455-458 "The filtering algorithm of biological-induced vertical velocities can be adjusted to DVMs, or to any region of the world ocean, coastal as offshore. Then the study of the remaining physics-driven vertical velocities will enhance our understanding of fine-scale oceanic processes, which have real consequences on how nutrients and other biogeochemical tracers get transported throughout the ocean."

The manuscript is readable and informative, but some syntax, grammar, and phrasing issues reduce its clarity. A careful proofreading of the manuscript should eliminate the most glaring errors, including the following:

All these corrections and suggestions have been taken into account.

Line 1: challenge of measuring

Changed.

Lines 7 and 139: three-dimensional

Changed line 7 and new line 144.

Line 15: most ocean dynamic processes

New line: 17.

Line 16: one of the most complex aspects

New lines: 17-18.

Line 16: usually of several orders

New line: 18.

Line 41: including in situ observations

New line 43.

Line 75: These forcings and their impact on the oceanic circulation have been studied

New lines: 76-77.

Line 135: offshore of Marseille on the border of the Gulf of Lion shelf

New line: 141.

Line 146: with the initial purpose of measuring

New line: 153. Rephrased to "The initial purpose was to measure the Northern Current intrusions on the continental shelf of the Gulf of Lion (Barrier et al., 2016)."

Line 152: thickness of the blanking

New line: 157.

Line 156: than that calculated

New line: 161.

Line 160: between 6 a.m. and 6 p.m.

New line: 170.

Line 173: to a 30 s temporal resolution

New line:183.

Line 175: from 78.9 to 3 m.

New line 184.

Line 262: filtered out from the W dataset,

New line 285.

Line 391: this work shows

New line: 438.

Line 397: "SUCH measurements" (unclear reference)

Indeed, thanks for pointing it. We rephrased it to: "W measured with JULIO exhibited strong intensity values for both upwellings and downwellings." new line 444.

Thank you again, your help has been highly appreciated.

---

## Author Response (AR1)

Estimating oceanic physics-driven vertical velocities in a wind-influenced coastal environment

Maxime Arnaud, Anne Petrenko, Jean-Luc Fuda, Caroline Comby, Anthony Bosse, Yann Ourmières, and Stéphanie Barrillon

**Letter to the editor**

Dear Editor,

We would like to thank you for having found rapidly two referees for our manuscript. We have carefully addressed all the comments raised by the referees, as well as by Bruno Blanke, and hope that this revised manuscript is found suitable for publication in Ocean Science.

We decided to add the latest JULIO time series (from July 12, 2023 to May 22, 2024) in our revised manuscript as we obtained the data during the review process. We think that it brings valuable information on our work as it adds  $\sim$  20% of data compared to the previous dataset in the original version of the paper.

Yours sincerely,

Maxime Arnaud,

on behalf of all co-authors.

---

## Author Response (AR2)

**Author's response to editor**

Maxime Arnaud1, Anne Petrenko1, Jean-Luc Fuda1, Caroline Comby1, Anthony Bosse1,

Yann Ourmières1, and Stéphanie Barrillon1

1Aix Marseille Univ., Université de Toulon, CNRS, MIO, 13288, Marseille, France

Corresponding author: <a href="maxime.arnaud@mio.osupytheas.fr">maxime.arnaud@mio.osupytheas.fr</a>

Dear Editor,

First, we would like to thank you for handling this paper and for your useful corrections.

We applied your suggestions and tried to enhance the paper flow by searching for similar technical issues. We also added the data availability section for JULIO with a link to our data assets.

Finally, about author's affiliation: this is the official affiliation given by our laboratory (MIO) which includes all of these components.

Kind regards,

Maxime Arnaud